# TESTING ROBUSTNESS AGAINST UNFORESEEN ADVERSARIES

## ABSTRACT

Most existing adversarial defenses only measure robustness to $L_p$ adversarial attacks. Not only are adversaries unlikely to exclusively create small $L_p$ perturbations, adversaries are unlikely to remain fixed. Adversaries adapt and evolve their attacks; hence adversarial defenses must be robust to a broad range of *unforeseen attacks*. We address this discrepancy between research and reality by proposing a new evaluation framework called ImageNet-UA. Our framework enables the research community to test ImageNet model robustness against attacks not encountered during training. To create ImageNet-UA's diverse attack suite, we introduce a total of four novel adversarial attacks. We also demonstrate that, in comparison to ImageNet-UA, prevailing $L_\infty$ robustness assessments give a narrow account of adversarial robustness. By evaluating current defenses with ImageNet-UA, we find they provide little robustness to unforeseen attacks. We hope the greater variety and realism of ImageNet-UA enables development of more robust defenses which can generalize beyond attacks seen during training.

## 1 INTRODUCTION

Neural networks perform well on many datasets (He et al., 2016) yet can be consistently fooled by minor adversarial distortions (Goodfellow et al., 2014). The research community has responded by quantifying and developing adversarial defenses against such attacks (Madry et al., 2017), but these defenses and metrics have two key limitations.

First, the vast majority of existing defenses exclusively defend against and quantify robustness to $L_p$-constrained attacks (Madry et al., 2017; Cohen et al., 2019; Raff et al., 2019; Xie et al., 2018). Though real-world adversaries are not $L_p$ constrained (Gilmer et al., 2018) and can attack with diverse distortions (Brown et al., 2017; Sharif et al., 2019), the literature largely ignores this and evaluates against the $L_p$ adversaries already seen during training (Madry et al., 2017; Xie et al., 2018), resulting in optimistic robustness assessments. The attacks outside the $L_p$ threat model that have been proposed (Song et al., 2018; Qiu et al., 2019; Engstrom et al., 2017; Evtimov et al., 2017; Sharif et al., 2016) are not intended for general defense evaluation and suffer from narrow dataset applicability, difficulty of optimization, or fragility of auxiliary generative models.

Second, existing defenses assume that attacks are known in advance (Goodfellow, 2019) and use knowledge of their explicit form during training (Madry et al., 2017). In practice, adversaries can deploy *unforeseen attacks* not known to the defense creator. For example, online advertisers use attacks such as perturbed pixels in ads to defeat ad blockers trained only on the previous generation of ads in an ever-escalating arms race (Tramèr et al., 2018). However, current evaluation setups implicitly assume that attacks encountered at test-time are the same as those seen at train-time, which is unrealistic. The reality that future attacks are unlike those encountered during training is akin to a train-test distribution mismatch—a problem studied outside of adversarial robustness (Recht et al., 2019; Hendrycks & Dietterich, 2019)—but now brought to the adversarial setting.

The present work addresses these limitations by proposing an evaluation framework ImageNet-UA to measure robustness against unforeseen attacks. ImageNet-UA assesses a defense which may have been created with knowledge of the commonly used $L_\infty$ or $L_2$ attacks with six diverse attacks (four of which are novel) distinct from $L_\infty$ or $L_2$. We intend these attacks to be used at *test-time* only and not during training. Performing well on ImageNet-UA thus demonstrates generalization to a diverse set of distortions not seen during defense creation. While ImageNet-UA

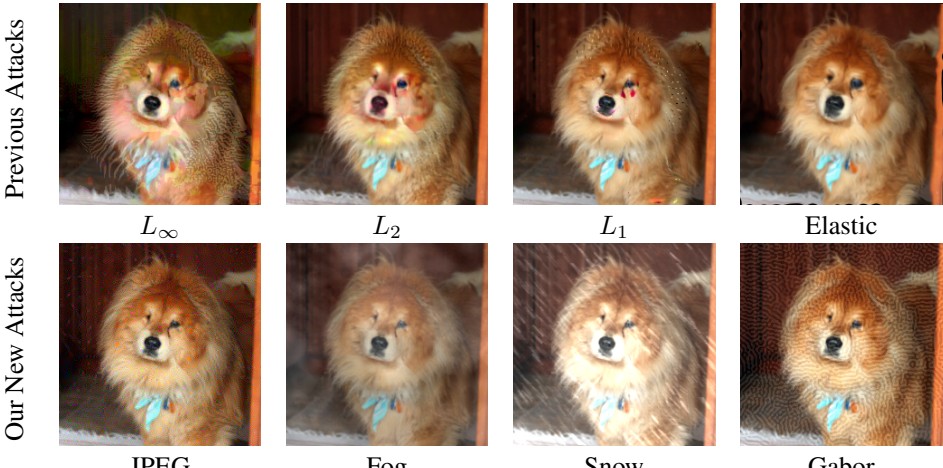

Figure 1: Adversarially distorted chow chow dog images created with old attacks and our new attacks. The JPEG, Fog, Snow, and Gabor adversarial attacks are visually distinct from previous attacks, result in distortions which do not obey a small $L_p$ norm constraint, and serve as unforeseen attacks for the ImageNet-UA attack suite.

does not provide an exhaustive guarantee over all conceivable attacks, it evaluates over a diverse unforeseen test distribution similar to those used successfully in other studies of distributional shift (Rajpurkar et al., 2018; Hendrycks & Dietterich, 2019; Recht et al., 2019). ImageNet-UA works for ImageNet models and can be easily used with our code available at https://github.com/anon-submission-2020/anon-submission-2020.

Designing ImageNet-UA requires new attacks that are strong and varied, since real-world attacks are diverse in structure. To meet this challenge, we contribute four novel and diverse adversarial attacks which are easily optimized. Our new attacks produce distortions with occlusions, spatial similarity, and simulated weather, all of which are absent in previous attacks. Performing well on ImageNet-UA thus demonstrates that a defense generalizes to a diverse set of distortions distinct from the commonly used $L_\infty$ or $L_2$.

With ImageNet-UA, we show weaknesses in existing evaluation practices and defenses through a study of 8 attacks against 48 models adversarially trained on ImageNet-100, a 100-class subset of ImageNet. While most adversarial robustness evaluations use only $L_\infty$ attacks, ImageNet-UA reveals that models with high $L_\infty$ attack robustness can remain susceptible to other attacks. Thus, $L_\infty$ evaluations are a narrow measure of robustness, even though much of the literature treats this evaluation as comprehensive (Madry et al., 2017; Qian & Wegman, 2019; Schott et al., 2019; Zhang et al., 2019). We address this deficiency by using the novel attacks in ImageNet-UA to evaluate robustness to a more diverse set of unforeseen attacks. Our results demonstrate that $L_\infty$ adversarial training, the current state-of-the-art defense, has limited generalization to unforeseen adversaries, and is not easily improved by training against more attacks. This adds to the evidence that achieving robustness against a few train-time attacks is insufficient to impart robustness to unforeseen test-time attacks (Jacobsen et al., 2019; Jordan et al., 2019; Tramèr & Boneh, 2019).

In summary, we propose the framework ImageNet-UA to measure robustness to a diverse set of attacks, made possible by our four new adversarial attacks. Since existing defenses scale poorly to multiple attacks (Jordan et al., 2019; Tramèr & Boneh, 2019), finding defense techniques which generalize to unforeseen attacks is crucial to create robust models. We suggest ImageNet-UA as a way to measure progress towards this goal.

## 2   RELATED WORK

Adversarial robustness is notoriously difficult to correctly evaluate (Papernot et al., 2017; Athalye et al., 2018a). To that end, Carlini et al. (2019a) provide extensive guidance for sound adversarial robustness evaluation. By measuring attack success rates across several distortion sizes and using a broader threat model with diverse differentiable attacks, ImageNet-UA has several of their recommendations built-in, while greatly expanding the set of attacks over previous work on evaluation.

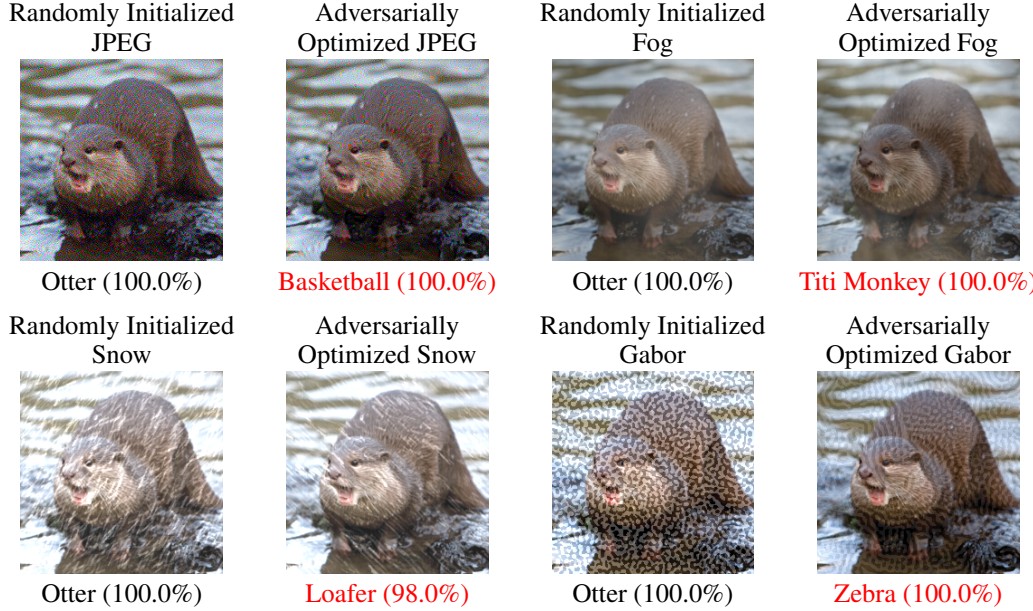

Figure 2: Randomly sampled distortions and adversarially optimized distortions from our new attacks, targeted to the target class in red. Stochastic average-case versions of our attacks affect classifiers minimally, while adversarial versions are optimized to reveal high-confidence errors. The snowflakes in Snow decrease in intensity after optimization, demonstrating that lighter adversarial snowflakes are more effective than heavy random snowfall at uncovering model weaknesses.

We are only aware of a few prior works which evaluate on unforeseen attacks in specific limited circumstances. Wu et al. (2020) evaluate against physically-realizable attacks from Evtimov et al. (2017) and Sharif et al. (2016), though this limits the threat model to occlusion attacks on narrow datasets. Outside of vision, Pierazzi et al. (2020) proposes constraining attacks by a more diverse set of problem-space constraints in diverse domains such as text and malware or source code generation; however, even in this framework, analytically enumerating all such constraints is impossible.

Within vision, prior attacks outside the $L_p$ threat model exist, but they lack the general applicability and fast optimization of ours. Song et al. (2018) and Qiu et al. (2019) attack using variational autoencoders and StarGANs, respectively, resulting in weaker attacks which require simple image distributions suitable for VAEs and GANs. Engstrom et al. (2017) apply Euclidean transformations determined by brute-force search. Zhao et al. (2019) use perceptual color distances to align human perception and $L_2$ perturbations. Evtimov et al. (2017) and Sharif et al. (2016) attack stop signs and face-recognition systems with carefully placed patches or modified eyeglass frames, requiring physical object creation and applying only to specific image types.

## 3 NEW ATTACKS FOR A BROADER THREAT MODEL

There are few diverse, easily optimizable, plug-and-play adversarial attacks in the current literature; outside of Elastic (Xiao et al., 2018), most are $L_p$ attacks such as $L_\infty$ (Goodfellow et al., 2014), $L_2$ (Szegedy et al., 2013; Carlini & Wagner, 2017), $L_1$ (Chen et al., 2018). We rectify this deficiency with four novel adversarial attacks: JPEG, Fog, Snow, and Gabor. Our attacks are differentiable and fast, while optimizing over enough parameters to be strong. We show example adversarial images in Figure 1 and compare stochastic and adversarial distortions in Figure 2.

Our novel attacks provide a range of *test-time* adversaries visually and semantically distinct from $L_\infty$ and $L_2$ attacks. Namely, they cause distortions with large $L_\infty$ and $L_2$ norm, but result in images that are perceptually close to the original. These attacks are intended as unforeseen attacks not used during training, allowing them to evaluate whether a defense can generalize from $L_\infty$ or $L_2$ to a more varied set of distortions than current evaluations. Though our attacks are not exhaustive, performing well against them already demonstrates robustness to occlusion, spatial similarity, and simulated weather, which are absent from previous evaluations.

Our attacks create an adversarial image $x'$ from a clean image $x$ with true label $y$. Let model $f$ map images to a softmax distribution, and let $\ell(f(x), y)$ be the cross-entropy loss. Given a target class $y' \neq y$, our attacks attempt to find a valid image $x'$ such that (1) the attacked image $x'$ is obtained by applying a distortion (of size controlled by a parameter $\varepsilon$) to $x$, and (2) the loss $\ell(f(x'), y')$ is minimized. An unforeseen adversarial attack is a white- or black-box adversarial attack unknown to the defense designer which does not change the true label of $x$ according to an oracle or human.

### 3.1 FOUR NEW UNFORESEEN ATTACKS

**JPEG.** JPEG applies perturbations in a JPEG-encoded space of compressed images rather than raw pixel space. More precisely, JPEG compression is a linear transform JPEG which applies color-space conversion, the discrete cosine transform, and then quantization. Our JPEG attack imposes the $L_\infty$-constraint

$$\|\mathsf{JPEG}(x) - \mathsf{JPEG}(x')\|_\infty \leq \varepsilon$$

on the attacked image $x'$. We optimize $z = \mathsf{JPEG}(x')$ under this constraint to find an adversarial perturbation in the resulting frequency space. The perturbed frequency coefficients are quantized, and we then apply a right-inverse of JPEG to obtain the attacked image $x'$ in pixel space. We use ideas from Shin & Song (2017) to make this differentiable. The resulting attack is conspicuously distinct from $L_p$ attacks.

**Fog.** Fog simulates worst-case weather conditions. Robustness to adverse weather is a safety critical priority for autonomous vehicles, and Figure 2 shows Fog provides a more rigorous stress-test than stochastic fog (Hendrycks & Dietterich, 2019). Fog creates adversarial fog-like occlusions by adversarially optimizing parameters in the diamond-square algorithm (Fournier et al., 1982) typically used to render stochastic fog effects.

This algorithm starts with random perturbations to the four corner pixels of the image. At step $t$, it iteratively perturbs pixels at the centers of squares and diamonds formed by those pixels perturbed at step $t-1$. The perturbation of a step $t$ pixel is the average of the neighboring step $t-1$ perturbations plus a parameter value which we adversarially optimize. We continue this process until all pixels have been perturbed; the outcome is a fog-like distortion to the original image.

**Snow.** Snow simulates snowfall with occlusions of randomly located small image regions representing snowflakes. Because the distortions caused by snowflakes are not differentiable in their locations, we instead place occlusions representing snowflakes at randomly chosen locations and orientations and adversarially optimize their intensities. This choice results in a fast, differentiable, and strong attack. Compared to synthetic stochastic snow (Hendrycks & Dietterich, 2019), our adversarial snow is faster and includes snowflakes at differing angles. Figure 2 shows adversarial snow exposes model weaknesses more effectively than the easier stochastic, average-case snow.

**Gabor.** Gabor spatially occludes the image with visually diverse Gabor noise Lagae et al. (2009). Gabor noise is a form of band-limited anisotropic procedural noise which convolves a parameter mask with a *Gabor kernel* which is a product of a Gaussian kernel and a harmonic kernel. We choose the Gabor kernel randomly and adversarially optimize the parameters of the mask starting from a sparse initialization. We apply spectral variance normalization (Co et al., 2019) to the resulting distortion and add it to the input image to create the attack.

### 3.2 IMPROVING EXISTING ATTACKS

Elastic modifies the attack of Xiao et al. (2018); it warps the image by distortions $x' = \mathsf{Flow}(x, V)$, where $V : \{1, \ldots, 224\}^2 \rightarrow \mathbb{R}^2$ is a vector field on pixel space, and Flow sets the value of pixel $(i, j)$ to the bilinearly interpolated original value at $(i, j) + V(i, j)$. We construct $V$ by smoothing a vector field $W$ by a Gaussian kernel (size $25 \times 25$, $\sigma \approx 3$ for a $224 \times 224$ image) and optimize $W$ under $\|W(i, j)\|_\infty \leq \varepsilon$ for all $i, j$. The resulting attack is suitable for large-scale images.

The other three attacks are $L_1, L_2, L_\infty$ attacks, but we improve the $L_1$ attack. For $L_\infty$ and $L_2$ constraints, we use randomly-initialized projected gradient descent (PGD), which applies gradient descent and projection to the $L_\infty$ and $L_2$ balls (Madry et al., 2017). Projection is difficult for $L_1$, and previous $L_1$ attacks rely on computationally intensive methods for it (Chen et al., 2018; Tramèr & Boneh, 2019). We replace PGD with the Frank-Wolfe algorithm (Frank & Wolfe, 1956), which

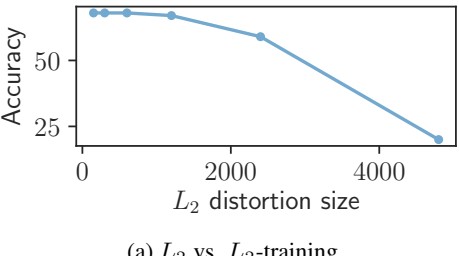 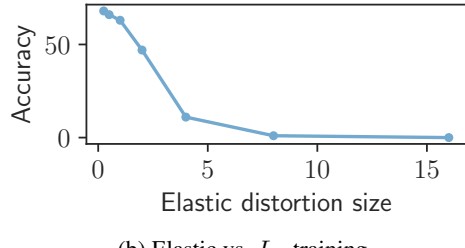

(a) $L_2$ vs. $L_2$-training          (b) Elastic vs. $L_2$-training

Figure 3: Accuracies of $L_2$ and Elastic attacks at different distortion sizes against a ResNet-50 model adversarially trained against $L_2$ at $\varepsilon = 9600$ on ImageNet-100. At small distortion sizes, the model appears to defend well against Elastic, but large distortion sizes reveal that robustness does not transfer from $L_2$ to Elastic.

optimizes a linear function instead of projecting at each step (pseudocode in Appendix D). This makes our $L_1$ attack more principled than previous implementations.

## 4  ImageNet-UA: MEASURING ROBUSTNESS TO UNFORESEEN ATTACKS

We propose the framework ImageNet-UA and its CIFAR-10 analogue CIFAR-10-UA to measure and summarize model robustness while fulfilling the following desiderata: (1) defenses should be evaluated against a broad threat model through a diverse set of attacks, (2) defenses should exhibit generalization to attacks not exactly identical to train-time attacks, and (3) the range of distortion sizes used for an attack must be wide enough to avoid misleading conclusions caused by overly weak or strong versions of that attack (Figure 3).

The ImageNet-UA evaluation framework aggregates robustness information into a single measure, the mean Unforeseen Adversarial Robustness (mUAR). The mUAR is an average over six different attacks of the Unforeseen Adversarial Robustness (UAR), a metric which assesses the robustness of a defense against a specific attack by using a wide range of distortion sizes. UAR is normalized using a measure of attack strength, the ATA, which we now define.

**Adversarial Training Accuracy (ATA).** The Adversarial Training Accuracy $\text{ATA}(A, \varepsilon)$ estimates the strength of an attack $A$ against adversarial training (Madry et al., 2017), one of the strongest known defense methods. For a distortion size $\varepsilon$, it is the best adversarial test accuracy against $A$ achieved by adversarial training against $A$. We allow a possibly different distortion size $\varepsilon'$ during training, since this can improves accuracy, and we choose a fixed architecture for each dataset.

For ImageNet-100, we choose ResNet-50 for the architecture, and for CIFAR-10 we choose ResNet-56. When evaluating a defense with architecture other than ResNet-50 or ResNet-56, we recommend using ATA values computed with these architectures to enable consistent comparison. To estimate $\text{ATA}(A, \varepsilon)$ in practice, we evaluate models adversarially trained against distortion size $\varepsilon'$ for $\varepsilon'$ in a large range (we describe this range at this section's end).

**UAR: Robustness Against a Single Attack.** The UAR, a building block for the mUAR, averages a model's robustness to a single attack over six distortion sizes $\varepsilon_1, \ldots, \varepsilon_6$ chosen for each attack (we describe the selection procedure at the end of this section). It is defined as

$$\text{UAR}(A) := 100 \times \frac{\sum_{k=1}^{6} \text{Acc}(A, \varepsilon_k, M)}{\sum_{k=1}^{6} \text{ATA}(A, \varepsilon_k)}, \tag{1}$$

where $\text{Acc}(A, \varepsilon_k, M)$ is the accuracy $\text{Acc}(A, \varepsilon_k, M)$ of a model $M$ after attack $A$ at distortion size $\varepsilon_k$. The normalization in (1) makes attacks of different strengths more commensurable in a stable way. We give values of $\text{ATA}(A, \varepsilon_k)$ and $\varepsilon_k$ for our attacks on ImageNet-100 and CIFAR-10 in Tables 4 and 5 (Appendix B), allowing computation of UAR of a defense against a single attack with six adversarial evaluations and no adversarial training.

**mUAR: Mean Unforeseen Attack Robustness.** We summarize a defense's performance on ImageNet-UA with the mean Unforeseen Attack Robustness (mUAR), an average of UAR scores

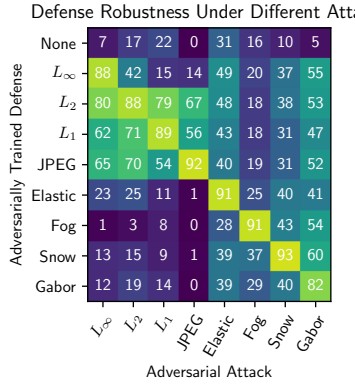

Figure 4: UAR for adv trained defenses (row) against attacks (col) on ImageNet-100. Defenses from $L_\infty$ to Gabor were trained with $\varepsilon = 32$, $4.8k$, $612k$, $2$, $16$, $8192$, $8$, and $1.6k$.

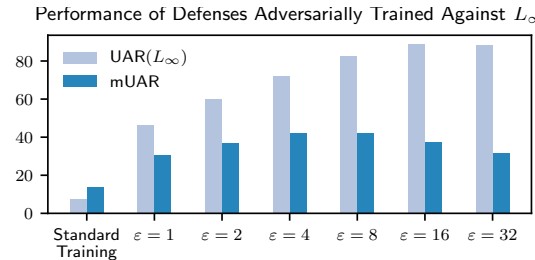

Figure 5: UAR($L_\infty$) and mUAR for $L_\infty$-trained models at different distortion sizes. Increasing distortion size in $L_\infty$-training improves UAR($L_\infty$) but hurts the mUAR, suggesting models heavily fit $L_\infty$ at the cost of generalization.

for the $L_1$, Elastic, JPEG, Fog, Snow, and Gabor attacks:

$$\mathsf{mUAR} := \frac{1}{6}\Big[\mathsf{UAR}(L_1) + \mathsf{UAR}(\text{Elastic}) + \mathsf{UAR}(\text{JPEG}) + \mathsf{UAR}(\text{Fog}) + \mathsf{UAR}(\text{Snow}) + \mathsf{UAR}(\text{Gabor})\Big].$$

Our measure mUAR estimates robustness to a broad threat model containing six unforeseen attacks at six distortion sizes each, meaning high mUAR requires generalization to several held-out attacks. In particular, it cannot be achieved by the common practice of engineering defenses to a single attack, which Figure 4 shows does not necessarily provide robustness to different attacks.

Our four novel attacks play a crucial role in mUAR by allowing us to estimate robustness to a sufficiently large set of adversarial attacks. As is customary when studying train-test mismatches and distributional shift, we advise against adversarially training with these six attacks when evaluating ImageNet-UA to preserve the validity of mUAR, though we encourage training with *other* attacks.

**Distortion Sizes.** We explain the $\varepsilon'$ values used to estimate ATA and the choice of $\varepsilon_1, \ldots, \varepsilon_6$ used to define UAR. This calibration of distortion sizes adjusts for the fact (Figure 3) that adversarial robustness against an attack may vary drastically with distortion size. Further, the relation between distortion size and attack strength varies between attacks, so too many or too few $\varepsilon_k$ values in a certain range may cause an attack to appear artificially strong or weak according to UAR.

We choose distortion sizes between $\varepsilon_{\min}$ and $\varepsilon_{\max}$ as follows. The minimum distortion size $\varepsilon_{\min}$ is the largest $\varepsilon$ for which the adversarial accuracy of an adversarially trained model at distortion size $\varepsilon$ is comparable to that of a model trained and evaluated on unattacked data (for ImageNet-100, within 3 of 87). The maximum distortion size $\varepsilon_{\max}$ is the smallest $\varepsilon$ which either reduces adversarial accuracy of an adversarially trained model at distortion size $\varepsilon$ below 25 or yields images confusing humans (adversarial accuracy can remain non-zero in this case).

As is typical in recent work on adversarial examples (Athalye et al., 2018b; Evtimov et al., 2017; Dong et al., 2019; Qin et al., 2019), our attacks can be perceptible at large distortion sizes. We make this choice to reflect perceptibility of attacks in real world threat models per Gilmer et al. (2018).

For ATA, we evaluate against models adversarially trained with $\varepsilon'$ increasing geometrically from $\varepsilon_{\min}$ to $\varepsilon_{\max}$ by factors of 2. We then choose $\varepsilon_k$ as follows: We compute ATA at $\varepsilon$ increasing geometrically from $\varepsilon_{\min}$ to $\varepsilon_{\max}$ by factors of 2 and take the size-6 subset whose ATA values have minimum $\ell_1$-distance to the ATA values of the $L_\infty$ attack in Table 4 (Appendix B.1). For example, for Gabor, $(\varepsilon_{\min}, \varepsilon_{\max}) = (6.25, 3200)$, so we compute ATAs at the 10 values $\varepsilon = 6.25, \ldots, 3200$. Viewing size-6 subsets of the ATAs as vectors with decreasing coordinates, we select $\varepsilon_k$ for Gabor corresponding to the vector with minimum $\ell_1$-distance to the ATA vector for $L_\infty$.

Table 1: Clean Accuracy, UAR, and mUAR scores for models adv trained against $L_\infty$ and $L_2$ attacks. $L_\infty$ training, the most popular defense, provides less robustness than $L_2$ training. Comparing the highest mUAR achieved to individual UAR values in Figure 4 indicates a large robustness gap.

| | Clean Accuracy | $L_\infty$ | $L_2$ | **mUAR** | | Clean Accuracy | $L_\infty$ | $L_2$ | **mUAR** |
|---|---|---|---|---|---|---|---|---|---|
| Normal Training | 86.7 | 7.3 | 17.2 | 14.0 | Normal Training | 86.7 | 7.3 | 17.2 | 14.0 |
| $L_\infty \, \varepsilon = 1$ | 86.2 | 46.4 | 54.2 | 30.7 | $L_2 \, \varepsilon = 150$ | 86.6 | 38.0 | 49.4 | 27.1 |
| $L_\infty \, \varepsilon = 2$ | 85.5 | 59.8 | 64.4 | 36.9 | $L_2 \, \varepsilon = 300$ | 85.9 | 49.7 | 60.1 | 33.3 |
| $L_\infty \, \varepsilon = 4$ | 83.9 | 72.1 | **73.6** | **42.3** | $L_2 \, \varepsilon = 600$ | 84.7 | 61.9 | 71.6 | 40.0 |
| $L_\infty \, \varepsilon = 8$ | 79.8 | 82.6 | 72.0 | 42.2 | $L_2 \, \varepsilon = 1200$ | 82.3 | 72.9 | 82.0 | 46.8 |
| $L_\infty \, \varepsilon = 16$ | 74.5 | **89.1** | 60.0 | 37.5 | $L_2 \, \varepsilon = 2400$ | 76.8 | 79.6 | **88.5** | **50.7** |
| $L_\infty \, \varepsilon = 32$ | 70.8 | 88.1 | 41.9 | 31.8 | $L_2 \, \varepsilon = 4800$ | 68.3 | **80.4** | 87.7 | 50.5 |

Table 2: Clean Accuracy, UAR, and mUAR scores for models jointly trained against $(L_\infty, L_2)$. Joint training does not provide much additional robustness.

| | Clean Accuracy | $L_\infty$ | $L_2$ | **mUAR** |
|---|---|---|---|---|
| $L_\infty \, \varepsilon = 1, L_2 \, \varepsilon = 300$ | 86.1 | 50.3 | 60.2 | 33.6 |
| $L_\infty \, \varepsilon = 2, L_2 \, \varepsilon = 600$ | 85.1 | 62.8 | 72.5 | 41.0 |
| $L_\infty \, \varepsilon = 4, L_2 \, \varepsilon = 1200$ | 81.3 | 72.9 | 81.2 | 46.9 |
| $L_\infty \, \varepsilon = 8, L_2 \, \varepsilon = 2400$ | 76.5 | 80.0 | 87.3 | 50.8 |
| $L_\infty \, \varepsilon = 16, L_2 \, \varepsilon = 4800$ | 68.4 | **81.5** | **87.9** | **50.9** |

## 5 New Insights From ImageNet-UA

We use ImageNet-UA to assess existing methods for adversarial defense and evaluation. First, ImageNet-UA reveals that $L_\infty$ trained defenses fail to generalize to different attacks, indicating substantial weakness in current $L_\infty$ adversarial robustness evaluation. We establish a baseline for ImageNet-UA using $L_2$ adversarial training which is difficult to improve upon by adversarial training alone. Finally, we show non-adversarially trained models can still improve robustness on ImageNet-UA over standard models and suggest this as a direction for further inquiry.

### 5.1 Experimental Setup

We adversarially train 48 models against the 8 attacks from Section 3 and evaluate against targeted attacks. We use the CIFAR-10 and ImageNet-100 datasets for ImageNet-UA and CIFAR-10-UA. ImageNet-100 is a 100-class subset of ImageNet-1K (Deng et al., 2009) containing every tenth class by WordNet ID order; we use a subset of ImageNet-1K due to the high compute cost of adversarial training. We use ResNet-56 for CIFAR-10 and ResNet-50 from `torchvision` for ImageNet-100 (He et al., 2016). We provide training hyperparameters in Appendix A.

To adversarially train against attack $A$, at each mini-batch we select a uniform random (incorrect) target class for each training image. For maximum distortion size $\varepsilon$, we apply targeted attack $A$ to the current model with distortion size $\varepsilon' \sim \text{Uniform}(0, \varepsilon)$ and take a SGD step using only the attacked images. Randomly scaling $\varepsilon'$ improves performance against smaller distortions.

We train on 10-step attacks for attacks other than Elastic, where we use 30 steps due to a harder optimization. For $L_p$, JPEG, and Elastic, we use step size $\varepsilon/\sqrt{\text{steps}}$; for Fog, Gabor, and Snow, we use step size $\sqrt{0.001/\text{steps}}$ because the latent space is independent of $\varepsilon$. These choices have optimal rates for non-smooth convex functions (Nemirovski & Yudin, 1978; 1983). We evaluate on 200-step targeted attacks with uniform random (incorrect) target, using more steps for evaluation than training per best practices (Carlini et al., 2019b).

Figure 4 summarizes ImageNet-100 results. Full results for ImageNet-100 and CIFAR-10 are in Appendix E and robustness checks to random seed and attack iterations are in Appendix F.

### 5.2 ImageNet-UA Reveals Weaknessess in $L_\infty$ Training and Testing

We use ImageNet-UA to reveal weaknesses in the common practices of $L_\infty$ robustness evaluation and $L_\infty$ adversarial training. We compute the mUAR and UAR($L_\infty$) for models trained against the $L_\infty$ attack with distortion size $\varepsilon$ and show results in Figure 5. For small $\varepsilon \leq 4$, mUAR and

Table 3: Non-adversarial defenses can noticeably improve ImageNet-UA performance. ResNeXt-101 (32×8d) + WSL is trained on approximately 1 billion images Mahajan et al. (2018). Stylized ImageNet is trained on a modification of ImageNet using style transfer Geirhos et al. (2019). Patch Gaussian augments using Gaussian distortions on small portions of the image Lopes et al. (2019). AugMix mixes simple random augmentations of the image Hendrycks et al. (2020). These results suggest that ImageNet-UA performance may be achieved through non-adversarial defenses.

| | Clean Acc. | $L_\infty$ | $L_2$ | $L_1$ | Elastic | JPEG | Fog | Snow | Gabor | **mUAR** |
|---|---|---|---|---|---|---|---|---|---|---|
| SqueezeNet | 84.1 | 5.2 | 11.2 | 14.9 | 25.9 | **1.9** | 20.1 | 9.8 | 4.4 | 12.8 |
| ResNeXt-101 (32×8d) | 95.9 | 2.5 | 5.5 | 20.7 | 26.5 | 1.8 | 14.1 | 12.4 | 5.3 | 13.4 |
| ResNeXt-101 (32×8d) + WSL | **97.1** | 3.0 | 5.7 | 28.3 | 29.4 | **1.9** | 26.2 | 20.3 | 8.0 | 19.0 |
| ResNet-18 | 91.6 | 2.7 | 8.2 | 13.5 | 22.6 | 1.8 | 20.3 | 9.5 | 4.2 | 12.0 |
| ResNet-50 | 94.2 | 2.7 | 6.6 | 20.1 | 24.9 | 1.8 | 15.8 | 11.9 | 4.9 | 13.2 |
| ResNet-50 + Stylized ImageNet | 94.6 | 2.9 | 7.4 | 22.8 | 26.0 | 1.8 | 16.2 | 12.5 | 8.1 | 14.6 |
| ResNet-50 + Patch Gaussian | 93.6 | 4.5 | 10.9 | 27.4 | 28.2 | 1.8 | 23.9 | 10.5 | 5.2 | 16.2 |
| ResNet-50 + AugMix | 95.1 | **6.1** | **13.4** | **34.3** | **38.8** | 1.8 | **28.6** | **24.7** | **11.1** | **23.2** |

UAR($L_\infty$) increase together with $\varepsilon$. For larger $\varepsilon \geq 8$, UAR($L_\infty$) continues to increase with $\varepsilon$, but the mUAR decreases, a fact which is not apparent from $L_\infty$ evaluation.

The decrease in mUAR while UAR($L_\infty$) increases suggests that $L_\infty$ adversarial training begins to heavily fit $L_\infty$ distortions at the expense of generalization at larger distortion sizes. Thus, while it is the most commonly used defense procedure, $L_\infty$ training may not lead to improvements on other attacks or to real-world robustness.

Worse, $L_\infty$ evaluation against $L_\infty$ adversarial training at higher distortions indicates higher robustness. In contrast, mUAR reveals that $L_\infty$ adversarial training at higher distortions in fact hurts robustness against a more diverse set of attacks. Thus, $L_\infty$ evaluation gives a misleading picture of robustness. This is particularly important because $L_\infty$ evaluation is the most ubiquitous measure of robustness in deep learning (Goodfellow et al., 2014; Madry et al., 2017; Xie et al., 2018).

### 5.3 LIMITS OF ADVERSARIAL TRAINING FOR ImageNet-UA

We establish a baseline on ImageNet-UA using $L_2$ adversarial training but show a significant performance gap even for more sophisticated existing adversarial training methods. To do so, we evaluate several adversarial training methods on ImageNet-UA and show results in Table 1.

Our results show that $L_2$ trained models outperform $L_\infty$ trained models and have significantly improved absolute performance, increasing mUAR from $14.0$ to $50.7$ compared to an undefended model. The individual UAR values in Figure 7 (Appendix E.1) improve substantially against all attacks other than Fog, including several (Elastic, Gabor, Snow) of extremely different nature to $L_2$.

This result suggests pushing adversarial training further by training against multiple attacks simultaneously via *joint adversarial training* (Jordan et al., 2019; Tramèr & Boneh, 2019) detailed in Appendix C. Table 2 shows that, despite using twice the compute of $L_2$ training, $(L_\infty, L_2)$ joint training only improves the mUAR from $50.7$ to $50.9$. We thus recommend $L_2$ training as a baseline for ImageNet-UA, though there is substantial room for improvement compared to the highest UARs against individual attacks in Figure 4, which are all above 80 and often above 90.

### 5.4 ImageNet-UA ROBUSTNESS THROUGH NON-ADVERSARIAL DEFENSES

We find that methods can improve robustness to unforeseen attacks without adversarial training. Table 3 shows mUAR for SqueezeNet (Iandola et al., 2017), ResNeXts (Xie et al., 2016), and ResNets. For ImageNet-1K models, we mask 900 logits to predict ImageNet-100 classes.

A popular defense against average case distortions (Hendrycks & Dietterich, 2019) is Stylized ImageNet (Geirhos et al., 2019), which modifies training images using image style transfer in hopes of making networks rely less on textural features. Table 3 shows it provides some improvement on ImageNet-UA. More recently, Lopes et al. (2019) propose to train against Gaussian noise applied to small image patches, improving the mUAR by 3% over the ResNet-50 baseline. The second largest mUAR improvement comes from training a ResNeXt on approximately 1 billion images (Mahajan et al., 2018). This three orders of magnitude increase in training data yields a 5.4% mUAR

increase over a vanilla ResNeXt baseline. Finally, Hendrycks et al. (2020) create AugMix, which randomly mixes stochastically generated augmentations. Although AugMix did not use random nor adversarial noise, it improves robustness to unforeseen attacks by 10%.

These results imply that defenses not relying on adversarial examples can improve ImageNet-UA performance. They indicate that training on more data only somewhat increases robustness on ImageNet-UA, unlike many other robustness benchmarks (Hendrycks & Dietterich, 2019; Hendrycks et al., 2019) where more data helps tremendously (Orhan, 2019). While models with lower clean accuracy (e.g., SqueezeNet and ResNet-18) have higher UAR($L_\infty$) and UAR($L_2$) than many other models, there is no clear difference in mUAR. Last, these non-adversarial defenses have minimal cost to accuracy on clean examples, unlike adversarial defenses. Much remains to explore, and we hope non-adversarial defenses will be a promising avenue toward adversarial robustness.

## 6 CONCLUSION

This work proposes a framework ImageNet-UA to evaluate robustness of a defense against *unforeseen attacks*. Because existing adversarial defense techniques do not scale to multiple attacks, developing models which can defend against attacks not seen at train-time is essential for robustness. Our results using ImageNet-UA show that the common practice of $L_\infty$ training and evaluation fails to achieve or measure this broader form of robustness. As a result, it can provide a misleading sense of robustness. By incorporating our 4 novel and strong adversarial attacks, ImageNet-UA enables evaluation on the diverse held-out attacks necessary to measure progress towards robustness more broadly.

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

## A    TRAINING HYPERPARAMETERS

For ImageNet-100, we trained on machines with 8 NVIDIA V100 GPUs using standard data augmentation (He et al., 2016). Following best practices for multi-GPU training (Goyal et al., 2017), we ran synchronized SGD for 90 epochs with batch size $32 \times 8$ and a learning rate schedule with 5 "warm-up" epochs and a decay at epochs 30, 60, and 80 by a factor of 10. Initial learning rate after warm-up was 0.1, momentum was 0.9, and weight decay was $10^{-4}$. For CIFAR-10, we trained on a single NVIDIA V100 GPU for 200 epochs with batch size 32, initial learning rate 0.1, momentum 0.9, and weight decay $10^{-4}$. We decayed the learning rate at epochs 100 and 150.

## B    CALIBRATION OF ImageNet-UA AND CIFAR-10-UA

### B.1    CALIBRATION FOR ImageNet-UA

Calibrated distortion sizes and ATA values are in Table 4.

### B.2    CALIBRATION FOR CIFAR-10-UA

The $\varepsilon$ calibration procedure for CIFAR-10 was similar to that used for ImageNet-100. We started with small $\varepsilon_{\min}$ values and increased $\varepsilon$ geometrically with ratio 2 until adversarial accuracy of an adversarially trained model dropped below 40. Note that this threshold is higher for CIFAR-10 than ImageNet-100 because there are fewer classes. The resulting ATA values for CIFAR-10 are shown in Table 5.

## C    JOINT ADVERSARIAL TRAINING

Our joint adversarial training procedure for two attacks $A$ and $A'$ is as follows. At each training step, we compute the attacked image under both $A$ and $A'$ and backpropagate with respect to gradients induced by the image with greater loss. This corresponds to the "max" loss of Tramèr & Boneh (2019). We train ResNet-50 models for $(L_\infty, L_2)$, $(L_\infty, L_1)$, and $(L_\infty, \text{Elastic})$ on ImageNet-100.

Table 6 shows training against $(L_\infty, L_1)$ is worse than training against $L_1$ at the same distortion size and performs particularly poorly at large distortion sizes. Table 7 shows joint training against

Table 4: Calibrated distortion sizes and ATA values for different distortion types on ImageNet-100.

| Attack | $\varepsilon_1$ | $\varepsilon_2$ | $\varepsilon_3$ | $\varepsilon_4$ | $\varepsilon_5$ | $\varepsilon_6$ | $\text{ATA}_1$ | $\text{ATA}_2$ | $\text{ATA}_3$ | $\text{ATA}_4$ | $\text{ATA}_5$ | $\text{ATA}_6$ |
|---|---|---|---|---|---|---|---|---|---|---|---|---|
| $L_\infty$ | 1 | 2 | 4 | 8 | 16 | 32 | 84.6 | 82.1 | 76.2 | 66.9 | 40.1 | 12.9 |
| $L_2$ | 150 | 300 | 600 | 1200 | 2400 | 4800 | 85.0 | 83.5 | 79.6 | 72.6 | 59.1 | 19.9 |
| $L_1$ | 9562.5 | 19125 | 76500 | 153000 | 306000 | 612000 | 84.4 | 82.7 | 76.3 | 68.9 | 56.4 | 36.1 |
| Elastic | 0.25 | 0.5 | 2 | 4 | 8 | 16 | 85.9 | 83.2 | 78.1 | 75.6 | 57.0 | 22.5 |
| JPEG | 0.062 | 0.125 | 0.250 | 0.500 | 1 | 2 | 85.0 | 83.2 | 79.3 | 72.8 | 34.8 | 1.1 |
| Fog | 128 | 256 | 512 | 2048 | 4096 | 8192 | 85.8 | 83.8 | 79.0 | 68.4 | 67.9 | 64.7 |
| Snow | 0.0625 | 0.125 | 0.25 | 2 | 4 | 8 | 84.0 | 81.1 | 77.7 | 65.6 | 59.5 | 41.2 |
| Gabor | 6.25 | 12.5 | 25 | 400 | 800 | 1600 | 84.0 | 79.8 | 79.8 | 66.2 | 44.7 | 14.6 |

Table 5: Calibrated distortion sizes and ATA values for ResNet-56 on CIFAR-10

| Attack | $\varepsilon_1$ | $\varepsilon_2$ | $\varepsilon_3$ | $\varepsilon_4$ | $\varepsilon_5$ | $\varepsilon_6$ | $\text{ATA}_1$ | $\text{ATA}_2$ | $\text{ATA}_3$ | $\text{ATA}_4$ | $\text{ATA}_5$ | $\text{ATA}_6$ |
|---|---|---|---|---|---|---|---|---|---|---|---|---|
| $L_\infty$ | 1 | 2 | 4 | 8 | 16 | 32 | 91.0 | 87.8 | 81.6 | 71.3 | 46.5 | 23.1 |
| $L_2$ | 40 | 80 | 160 | 320 | 640 | 2560 | 90.1 | 86.4 | 79.6 | 67.3 | 49.9 | 17.3 |
| $L_1$ | 195 | 390 | 780 | 1560 | 6240 | 24960 | 92.2 | 90.0 | 83.2 | 73.8 | 47.4 | 35.3 |
| JPEG | 0.03125 | 0.0625 | 0.125 | 0.25 | 0.5 | 1 | 89.7 | 87.0 | 83.1 | 78.6 | 69.7 | 35.4 |
| Elastic | 0.125 | 0.25 | 0.5 | 1 | 2 | 8 | 87.4 | 81.3 | 72.1 | 58.2 | 45.4 | 27.8 |

Table 6: UAR scores for $L_1$-trained models and $(L_\infty, L_1)$-jointly trained models. At each distortion size, $L_1$-training performs better than joint training.

|  | $\mathrm{UAR}_{L_\infty}$ | $\mathrm{UAR}_{L_1}$ |
|---|---|---|
| $L_\infty\ \varepsilon = 2,\ L_1\ \varepsilon = 76500$ | 48 | 66 |
| $L_\infty\ \varepsilon = 4,\ L_1\ \varepsilon = 153000$ | **51** | **72** |
| $L_\infty\ \varepsilon = 8,\ L_1\ \varepsilon = 306000$ | 44 | 62 |
| $L_1\ \varepsilon = 76500$ | 50 | 70 |
| $L_1\ \varepsilon = 153000$ | 54 | 81 |
| $L_1\ \varepsilon = 306000$ | **59** | **87** |

Table 7: UAR scores for $L_\infty$- and Elastic-trained models and $(L_\infty,$ Elastic$)$-jointly trained models. No jointly trained model matches a Elastic-trained model on UAR vs. Elastic.

|  | $\mathrm{UAR}_{L_\infty}$ | $\mathrm{UAR}_{\mathrm{Elastic}}$ |
|---|---|---|
| $L_\infty\ \varepsilon = 4,$ Elastic $\varepsilon = 2$ | 68 | 63 |
| $L_\infty\ \varepsilon = 8,$ Elastic $\varepsilon = 4$ | 35 | **65** |
| $L_\infty\ \varepsilon = 16,$ Elastic $\varepsilon = 8$ | **69** | 43 |
| Elastic $\varepsilon = 2$ | **37** | 68 |
| Elastic $\varepsilon = 4$ | 36 | 81 |
| Elastic $\varepsilon = 8$ | 31 | **91** |

$(L_\infty,$ Elastic$)$ also performs poorly, never matching the UAR score of training against Elastic at moderate distortion size $(\varepsilon = 2)$.

## D   THE FRANK-WOLFE ALGORITHM

We chose to use the Frank-Wolfe algorithm for optimizing the $L_1$ attack, as Projected Gradient Descent would require projecting onto a truncated $L_1$ ball, which is a complicated operation. In contrast, Frank-Wolfe only requires optimizing linear functions $g^\top x$ over a truncated $L_1$ ball; this can be done by sorting coordinates by the magnitude of $g$ and moving the top $k$ coordinates to the boundary of their range (with $k$ chosen by binary search). This is detailed in Algorithm 1.

## E   FULL EVALUATION RESULTS

### E.1   FULL EVALUATION RESULTS AND ANALYSIS FOR IMAGENET-100

We show the full results of all adversarial attacks against all adversarial defenses for ImageNet-100 in Figure 6. These results also include $L_1$-JPEG and $L_2$-JPEG attacks, which are modifications of the JPEG attack applying $L_p$-constraints in the compressed JPEG space instead of $L_\infty$ constraints. Full UAR scores are provided for ImageNet-100 in Figure 7.

### E.2   FULL EVALUATION RESULTS AND ANALYSIS FOR CIFAR-10

We show the results of adversarial attacks and defenses for CIFAR-10 in Figure 8. We experienced difficulty training the $L_2$ and $L_1$ attacks at distortion sizes greater than those shown and have omitted those runs, which we believe may be related to the small size of CIFAR-10 images. Full UAR values for CIFAR-10 are shown in Figure 9.

## F   ROBUSTNESS OF OUR RESULTS

### F.1   REPLICATION

We replicated our results for the first three rows of Figure 6 with different random seeds to see the variation in our results. As shown in Figure 10, deviations in results are minor.

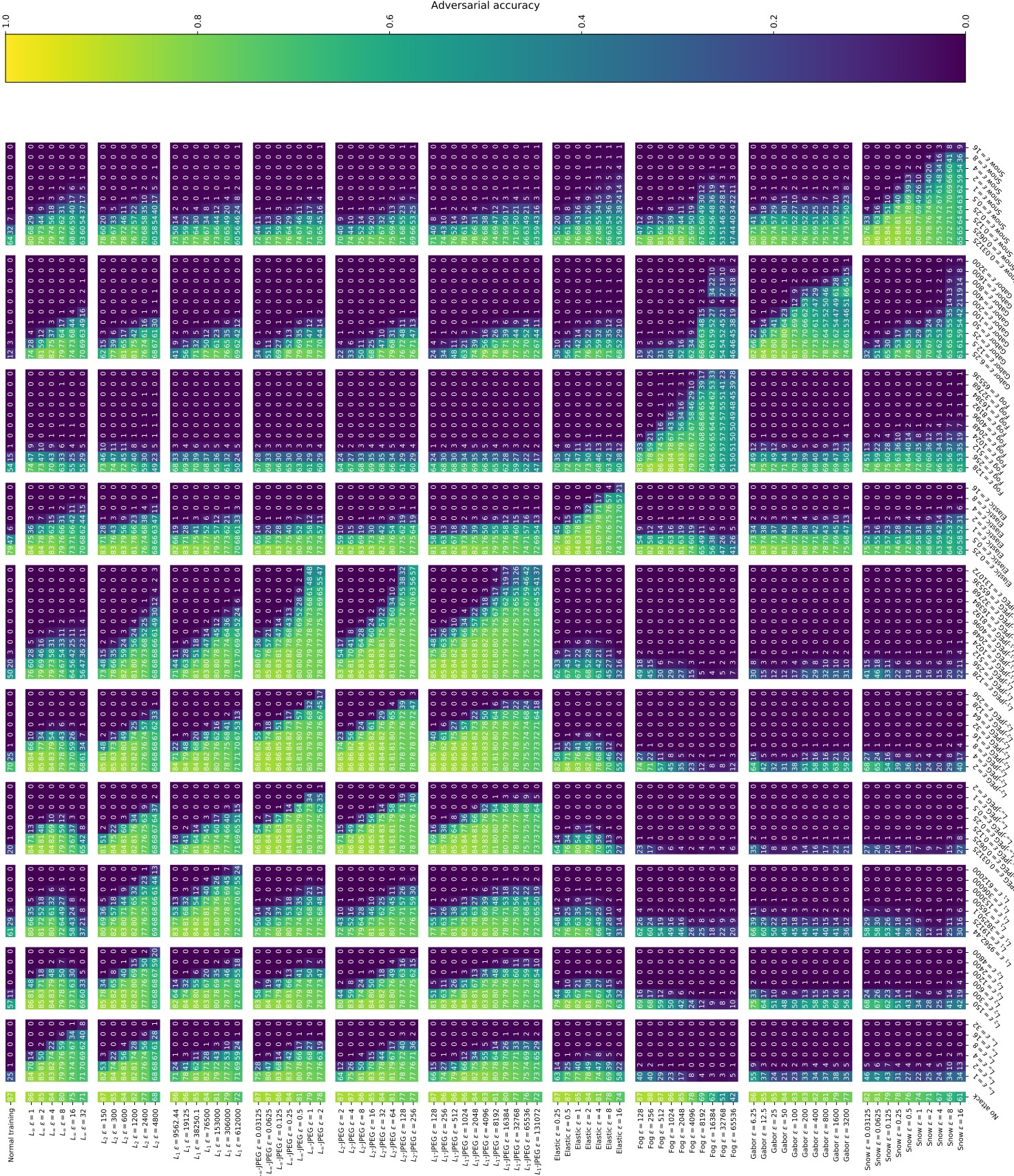

Figure 6: Accuracy of adversarial attack (column) against adversarially trained model (row) on ImageNet-100.

---

**Algorithm 1** Pseudocode for the Frank-Wolfe algorithm for the $L_1$ attack.

---

1: **Input:** function $f$, initial input $x \in [0,1]^d$, $L_1$ radius $\rho$, number of steps $T$.
2: **Output:** approximate maximizer $\bar{x}$ of $f$ over the truncated $L_1$ ball $B_1(\rho; x) \cap [0,1]^d$ centered at $x$.
3:
4: $x^{(0)} \leftarrow \text{RandomInit}(x)$ {Random initialization}
5: **for** $t = 1, \ldots, T$ **do**
6:     $g \leftarrow \nabla f(x^{(t-1)})$ {Obtain gradient}
7:     **for** $k = 1, \ldots, d$ **do**
8:         $s_k \leftarrow$ index of the coordinate of $g$ by with $k^{\text{th}}$ largest norm
9:     **end for**
10:     $S_k \leftarrow \{s_1, \ldots, s_k\}$.
11:
12:     {Compute move to boundary of $[0,1]$ for each coordinate.}
13:     **for** $i = 1, \ldots, d$ **do**
14:         **if** $g_i > 0$ **then**
15:             $b_i \leftarrow 1 - x_i$
16:         **else**
17:             $b_i \leftarrow -x_i$
18:         **end if**
19:     **end for**
20:     $M_k \leftarrow \sum_{i \in S_k} |b_i|$ {Compute $L_1$-perturbation of moving $k$ largest coordinates.}
21:     $k^* \leftarrow \max\{k \mid M_k \leq \rho\}$ {Choose largest $k$ satisfying $L_1$ constraint.}
22:
23:     {Compute $\hat{x}$ maximizing $g^\top x$ over the $L_1$ ball.}
24:     **for** $i = 1, \ldots, d$ **do**
25:         **if** $i \in S_{k^*}$ **then**
26:             $\hat{x}_i \leftarrow x_i + b_i$
27:         **else if** $i = s_{k^*+1}$ **then**
28:             $\hat{x}_i \leftarrow x_i + (\rho - M_{k^*})\,\text{sign}(g_i)$
29:         **else**
30:             $\hat{x}_i \leftarrow x_i$
31:         **end if**
32:     **end for**
33:     $x^{(t)} \leftarrow (1 - \frac{1}{t})x^{(t-1)} + \frac{1}{t}\hat{x}$ {Average $\hat{x}$ with previous iterates}
34: **end for**
35: $\bar{x} \leftarrow x^{(T)}$

---

## F.2 Convergence

We replicated the results in Figure 6 with 50 instead of 200 steps to see how the results changed based on the number of steps in the attack. As shown in Figure 11, the deviations are minor.

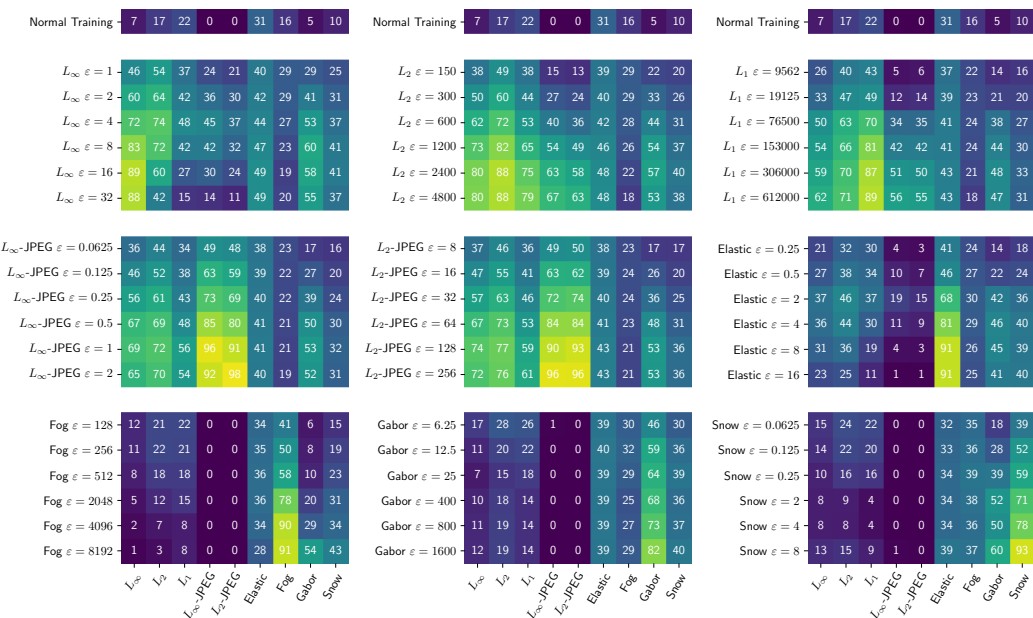

Figure 7: UAR scores for adv. trained defenses (rows) against distortion types (columns) for ImageNet-100.

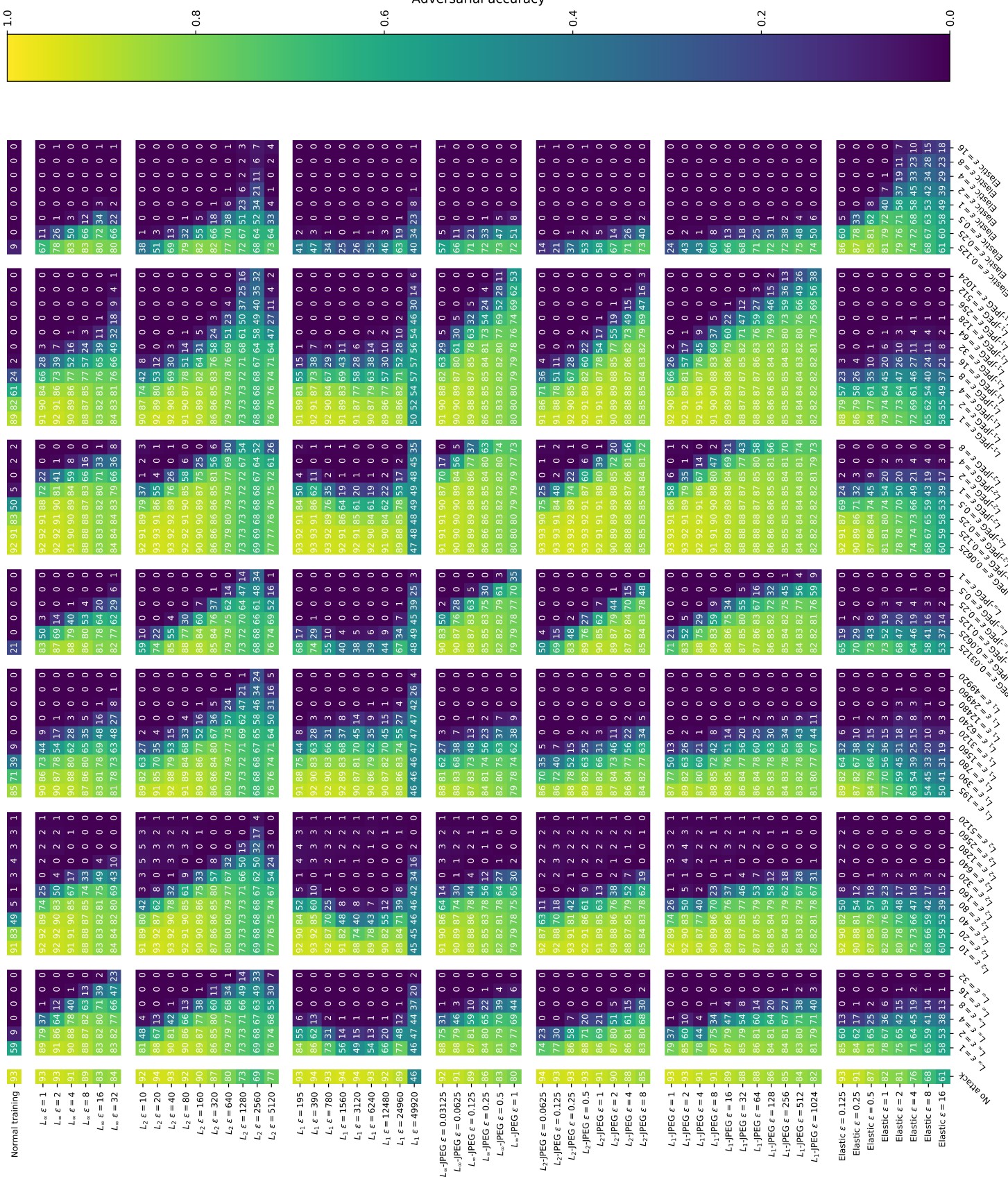

Figure 8: Accuracy of adversarial attack (column) against adversarially trained model (row) on CIFAR-10.

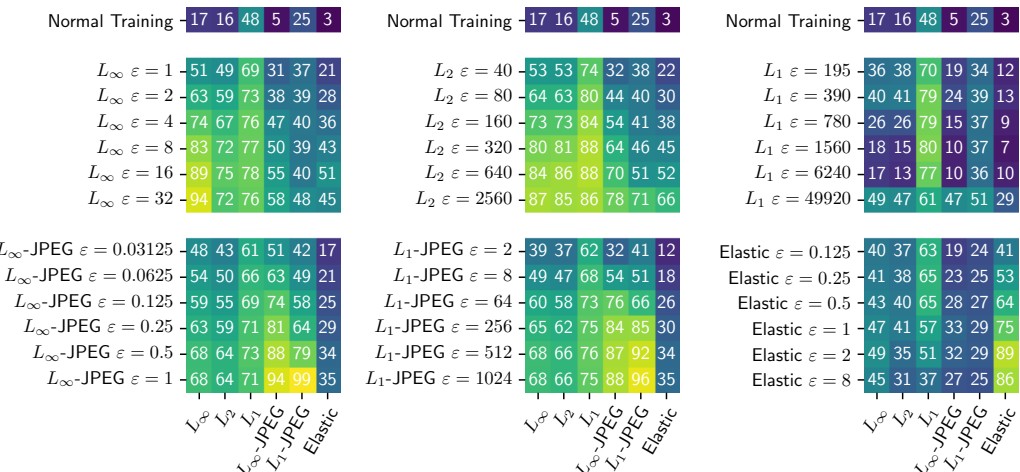

Figure 9: UAR scores on CIFAR-10. Displayed UAR scores are multiplied by 100 for clarity.

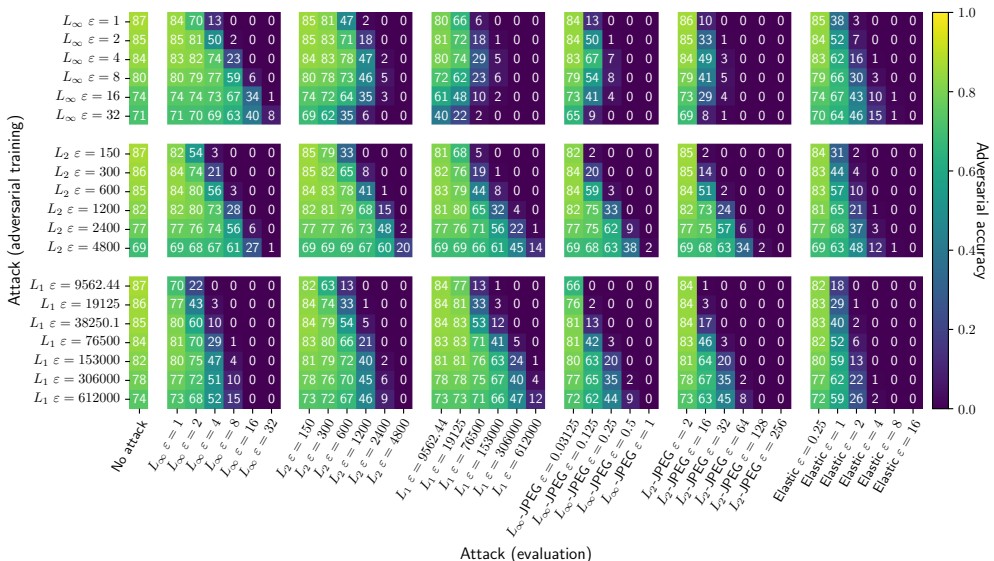

Figure 10: Replica of the first three block rows of Figure 6 with different random seeds. Deviations in results are minor.

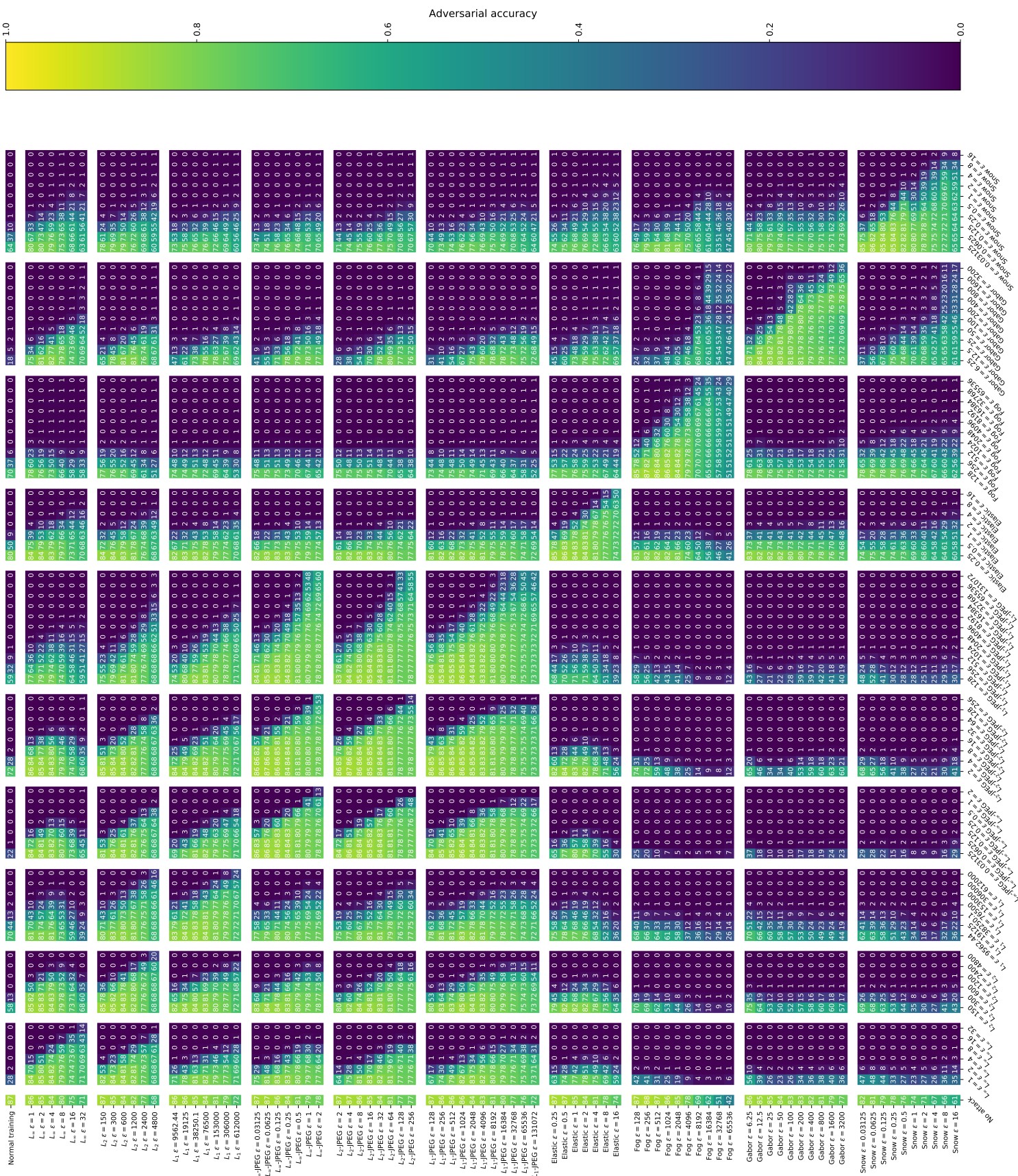

Figure 11: Replica of Figure 6 with 50 steps instead of 200 at evaluation time. Deviations in results are minor.

