# OpenReview forum: "Testing Robustness Against Unforeseen Adversaries"
_ICLR.cc/2021/Conference — Reject_

### Official Review · AnonReviewer2 · 2020-10-28
**An interesting idea**

**Rating:** 4
**Confidence:** 3

**Review:**

This paper presents the existence of several adversarial attacks that have meaningful visual concepts. Based on these attacks, the authors further develop a method to measure the robustness of a network or an adversarial defense.

Despite being conceptually interesting, this paper is working on the thing that many studies have done. The related work section is not on the point, in my opinion, the core problem (''unforeseen adversaries'') of this work should be ''adversarial attacks that are designed to break the commonly employed defenses'', but not ''adversarial attacks that are visually meaningful, and occasionally break the commonly employed defenses''. Therefore, the related work should focus on these studies that propose measures of robustness, and the experimental analysis should focus on doing a comparison of these measures, telling why the readers should adopt the ImageNet-UA and mUAR as their baseline, but not others.

On the other hand, the analysis of these newly developed is not sufficient, and the experimental results can not support the claims convincingly. For instance, in Table 3, the JPEG attack acquires the worst performance and the data augmentation methods do not help in all scenarios. Meanwhile, the Fog and Snow attacks are less effective than the $L_\inf$ and $L_2$ and benefit from more data. This implies they have different dynamics thus should not be used for mUAR homogeneously.

Anyway, I suggest the authors carefully revise the motivation and experimental analysis.

---

> ### Author Response · Authors · 2020-11-11
> **Clarification on setting of ImageNet-UA**
>
> We thank the reviewer for the review and comments.  We believe there is a crucial misunderstanding of the setting of our paper, which we seek to clarify in the next section.
>
> ## Comparison to other measures of robustness
> We disagree that our paper is working on the same problem as the many previous papers on measuring robustness.  Almost all such papers take a narrow definition of robustness to a specific type of adversary (e.g. with bounded L_inf or L_2 norm) and attempt to directly measure or produce proxies for it.  Such evaluations fail to consider attacks aside from L_inf or L_2 and reward defenses which overfit to these specific attacks.  Our results show that overfitting can degrade performance against other attacks, making these measures inappropriate for our unforeseen setting.
>
> To our knowledge, there is minimal prior work on evaluating defenses against unforeseen attacks.   We have compared our approach to that of Wu et. al. (2020), which is restricted to unforeseen physically realizable attacks, meaning it is not applicable for general comparison.  A major contribution of our work is to create a framework to make such evaluations possible for a more general problem, and ImageNet-UA is the only baseline we know of in the unforeseen setting.
>
> Though we do not claim it is exhaustive, we believe mUAR is a simple and useful summary statistic for measuring progress in robustness in the unforeseen setting.  In particular, obtaining a high mUAR without referencing the attacks in ImageNet-UA would provide evidence of generalization to at least 6 diverse unforeseen attacks.  To our knowledge, no other existing evaluation scheme allows defense creators to test their models in this way.
>
> We hope that the reviewer can reevaluate our paper from this viewpoint as a paper specifically on evaluation against unforeseen attacks rather than simply robustness to a single attack.  We believe that this problem has not been adequately addressed in the literature, and do not know of previous studies in this direction.  If the reviewer has other previous work in mind, we would love to learn of it and to see how it relates to our work. If this allays the reviewer’s concerns about the novelty and relevance of our work, we would respectfully ask the reviewer to consider improving the rating of this paper.
>
> ## Evaluation of attacks and usage in mUAR
> We believe the heterogeneous behavior of the attacks is one of the main benefits of our framework because it can demonstrate different aspects of defense behavior.  Because we did not see a pervasive trend in attack strength which extends across all defenses in our evaluations, we do not believe it is appropriate to reweight the attacks when constructing mUAR (The trends noted by the reviewer in Table 3 are often reversed when evaluating against adversarially trained models in Figure 7.)  Instead, we have followed other commonly used heterogeneous multi-task benchmarks (e.g. ImageNet-C, SuperGlue) in weighing them equally.
>
> More broadly, we agree that the behavior differences between the different attacks in ImageNet-UA are interesting and form a direction for future study.  Because of the number of attacks in ImageNet-UA, however, these differences are not so major as to affect the interpretation of our qualitative results on overfitting to L_inf and L_2 in Section 5 or other future applications of our framework to evaluate generalization of defenses to unforeseen attacks.
>
> Finally, we would like to point out that while evaluation against diverse attacks is commonly advocated, it is rare in practice because designing general, easily-optimized, and strong attacks is quite challenging and few are available.  We believe that enabling evaluation against such diverse attacks is a major contribution of our work.

---

### Official Review · AnonReviewer3 · 2020-10-28
**Good motivation but not unconvincing enough**

**Rating:** 5
**Confidence:** 3

**Review:**

This paper proposes four novel efficient adversarial attack methods beyond Lp threat models. Together with other two existing attack methods, these six attack methods combine as a framework to evaluate robustness of defenses against unforeseen attacks. In this framework, the novel measure is normalized with the performance of adversarial training. The experiments show that the Linf adversarially trained model may not lead to improvement of robustness against other threat models. It is expected that the framework could help test model robustness.

## Advantages

- Evaluating model robustness against unforeseen adversaries is an important problem.
- The four proposed adversarial attacks are differentiable and easy to use.

## Disadvantages

- The proposed measure mUAR is quite heuristic, e.g., the choice of the epsilon parameter.
- As a baseline framework, it is hard to say that the six adversarial attacks are sufficient to evaluate robustness against unforeseen adversaries. As a result, the claims seem not to be very convincing.
- In general, the result that adversarial training is not robust to unforeseen attacks is not surprising and has be been discussed a lot in literature.  It would be good to evaluate more existing defenses in order to derive more novel, interesting and inspiring insights.

---

> ### Author Response · Authors · 2020-11-11
> **Responses to review**
>
> We thank the reviewer for the review and comments, which we respond to in detail below.
>
> ## Heuristic nature of mUAR
> To measure progress through a benchmark, we must unavoidably make some arbitrary choices for precise values of parameters, which is acceptable as long as these choices do not affect qualitative trends in the results.  Many successful existing benchmarks make similarly arbitrary choices when evaluating on specific tasks (e.g. ImageNet-C, Hendrycks & Dietterich 2019), and constructing a summary statistic by averaging over several heterogeneous tasks in the same domain is also common (ImageNet-C, SuperGLUE, etc.).
>
> We chose parameters in mUAR to ensure they lie within a range where qualitative trends do not depend on their precise values, which we checked by evaluating over a range of epsilons, running extensive experiments with many adversarially trained models, and verifying convergence and replication of our training and evaluation procedure (Appendix F).  We believe this is a substantial improvement over existing adversarial robustness evaluations, which frequently compare defense robustness against a single attack at a single epsilon value.
>
> We hope this can address the referee’s concerns about our parameter choices in ImageNet-UA.  If there are concerns about the importance of other choices we made, we would be happy to elaborate in more detail about them.
>
> ## Is evaluating against six attacks sufficient to measure robustness?
> We believe there is a misunderstanding of our intended role for ImageNet-UA.  Before our work, it was quite difficult to evaluate unforeseen robustness against a sufficiently diverse set of attacks, as attacks used in general robustness evaluations must be general-purpose, fast-to-optimize, and strong, properties which required significant novelty to design for in the 4 new attacks in ImageNet-UA.
>
> As such, we view performing well on ImageNet-UA as a way for defenses to demonstrate generalization to at least the 6 diverse attacks in ImageNet-UA.  Even this level of unforeseen robustness was difficult to easily measure prior to this work, and we demonstrate that existing techniques are not close to achieving it.  Though we agree that ImageNet-UA is not an exhaustive measure of robustness to unforeseen attacks, the extreme diversity of plausible attacks in the unforeseen setting means it may be impossible to produce such a measure.  On the other hand, we believe that ImageNet-UA is a diverse enough evaluation to drive progress in this direction.
>
> We hope this correctly understands and allays the referee’s concerns about the usage of ImageNet-UA in evaluation.  We are happy to discuss this further, and if the reviewer agrees we would respectfully ask the reviewer to consider improving the rating of this paper.
>
> ## Novel insights
> Our primary contribution is to establish ImageNet-UA as a benchmark for measuring robustness to unforeseen adversaries and showing that no defense generalizes to these particular attacks. In addition to the lack of transfer (which we agree is known), we also show that defenses can _overfit_ to the training-time attack, meaning that adversarial training against one attack can _degrade_ adversarial robustness to other attacks.  This further demonstrates the need for evaluations with a variety of attacks, as provided by ImageNet-UA.

---

### Official Review · AnonReviewer1 · 2020-11-03
**The paper proposes a novel benchmark for evaluation of the model's robustness against unforeseen adversaries. The authors should provide implementation details of the novel attacks and make the source code available during the reviewing process.**

**Rating:** 5
**Confidence:** 5

**Review:**

**Update**: Thanks to the authors for addressing my comments and releasing the source code. Code is well-structured and easy to follow. It can be definitely used as a supplement for the robustness evaluation. However, judging by the implementation, the rain, snow, and fog attacks are too simplistic. From the paper, it is not clear how well these attacks approximate the respective type of perturbations in real-world scenarios. Therefore, the proposed set of 6 attacks is more the author's heuristic than something that can be used for the evaluation of computer vision systems, e.g. autonomous vehicles. Given all that, I decrease my score from 5 to 4.

###### Summary
The paper proposes a novel benchmark for the evaluation of the model's robustness against unforeseen adversaries. The authors suggested using 6 types of adversaries: JPEG, FOG, SNOW, and GABOR attack, L1, ELASTIC. Fully-differentiable variants of the above attacks are based on the previous work. The authors introduced the novel $l_1$-norm attack, which uses the Frank-Wolf algorithm to satisfy $l_1$-perturbation constraint. The authors introduced two metrics to compare defenses: robustness against a single unforeseen attack and mean unforeseen robustness. Experiments with adversarially trained models and non-adversarially trained models align with the previous work in the area.

###### Reasons for score:
I vote for a weak accept. The paper studies an important problem of the robustness of unforeseen adversaries. Introducing novel benchmarks beyond $l_{p}$-norm robustness is an important problem. The paper is clearly written and easy to follow. The evaluation results are extensive. However, implementation details of some attacks (SNOW, FOG, GABOR) are missing. JPEG attack is based on the previous work. It is not clear if the authors plan to release the benchmark suite to the reviewers. The most similar work in Hendrycks et al. 2019 uses a more diverse set of attacks.

###### Concerns:
- Novel differentiable attacks are based on previous work, e.g. differentiable JPEG and FOG. The algorithm and implementation details for SNOW and GABOR attacks are not provided. It is not clear how to reimplement these attacks.
- The paper is the most similar to Hendrycks & Dietterich 2019. The main difference is the use of differentiable attacks. However, previous work in Hendrycks uses a larger set of corruptions. This work includes only 5 corruptions from Hendrycks et al. 2019. Do the authors plan to include other types of attacks for the Imagenet-UA benchmark?
- It is not clear from the main text if the authors plan to release the benchmark source code during the reviewing process.

###### Minor comments:
- Comment that Elastic $l_1$ attack is based on heuristics might be incorrect. Elastic $l_1$ attack uses a proximity operator, a principal way to minimize $l_1$-norm.

---

> ### Author Response · Authors · 2020-11-11
> **Clarification that our code is open-sourced**
>
> We thank the reviewer for their review and comments.  The primary concern seems to be about the reproducibility and implementation details of our evaluation and attacks.  We believe there was a significant miscommunication about this, which we clarify below.
>
> We strongly believe in the importance of easy to use, open-source benchmarks, so we have [open sourced our code anonymously](https://github.com/anon-submission-2020/anon-submission-2020): https://github.com/anon-submission-2020/anon-submission-2020 (as linked in our original submission).  This provides full implementation details and scripts to easily reproduce our UAR and mUAR values on ImageNet-100 and CIFAR-10.
>
> As suggested, we will add further algorithmic details of the attacks in a forthcoming revision.  However, we believe that they are less informative than viewing the sample attacked images in our submission and the code we provide.  Namely, the main benefit of our attacks is that they produce images plausible to the eye, while the precise attack process involves details of image generation techniques which may be less essential.
>
> We hope that we have correctly understood the reviewer’s comments and that this allays the reviewer’s concerns about reproducibility of our attacks and benchmark. If the reviewer agrees we would respectfully ask the reviewer to consider improving the rating of this paper.
>
> ## Including other attacks in ImageNet-UA
> We have chosen not to include more attacks of this genre in ImageNet-UA for two reasons:
> 1. Because they optimize adversarially, our novel differentiable attacks require more compute to execute than the common corruptions of Hendrycks & Dietterich 2019 and are not fixed (i.e., depend on the defense method at hand). We have chosen the 6 attacks in ImageNet-UA to maintain diversity while fitting within computational budgets of academic researchers.
> 2. Turning the average-case common corruptions in Hendrycks & Dietterich 2019 into strong, easily-optimized attacks required choosing a sufficiently large space of parameters to adversarially optimize while keeping the optimization tractable.  In some cases (Snow) this required us to modify the generative process for the corruptions.  Of the corruptions in Hendrycks & Dietterich 2019 we did not select, some vary a small number of parameters and are not suitable (brightness, contrast, pixelate), while others we tried did not result in strong attacks due to optimization difficulties.
>
> Of course, it would be interesting to see other new general attacks of this nature, but for the reasons above, we believe that the current set is an appropriate choice for ImageNet-UA evaluation.
>
> ## Elastic L1
> Thanks for bringing this oversight to our attention.  We will update the discussion of the EAD L1 attack to reflect this.

---

### Official Review · AnonReviewer4 · 2020-11-08
**ICLR 2021 Conference Paper2332 AnonReviewer4 Review**

**Rating:** 5
**Confidence:** 4

**Review:**

## Summary

The authors propose ImageNet-UA, a framework to evaluate the
effectiveness of test-time unforeseen adversarial ML attacks. In this
particular context, the authors observe that attackers are not limited
to induce minimal Lp-norm perturbations and therefore it is important to
understand how robust existing defenses are against attacks that diverge
from these constraints while still being realizable. Within their
framework, the authors propose four novel adversarial attacks to
consolidate our understanding of test-time adversarial ML attacks.

## Strengths

+  Interesting intuition that test-time adversarial attacks are not all
necessarily driven by minimal perturbations
+  Attack-evaluation framework available to the community

## Weaknesses

-  Recent work explores the need for having a unified theoretical
   framework to reason about realizable attacks
-  Actual benefit of the approach

## Comments

This is an interesting paper that explores an avenue that is often
neglected in the context of test-time adversarial ML attacks. There is a
tendency on focusing on attacks that minimize the perturbation in a
given Lp-norm, when in fact there are several application domains for
which this constraint does not necessarily matter. Here, in the context
of realizable or problem-space attacks, one is required to reformulate
the problem, which may include other constraints attacks must satisfy.

As the authors admit, ImageNet-UA is not exhaustive although it tries to
evaluate the robustness of defenses over diverse unforeseen test
(adversarial) distributions. Under this premise, I somehow fail to see
the real benefit of such a framework, unless the authors show such
additional attacks comprise widely-used transformations that should
therefore become part of the threat model of any adversarial work in the
problem space. (In addition, most of the attacks seem to be a marginal
improvement of existing ones.) Conversely, a theoretical reformulation of the
problem-space would have been perhaps more useful. That would have
allowed one to reason about properties or set of transformations one
could or should consider, along with other constraints, as outlined in
[1].

I also wonder what's the rationale behind the mUAR metric. It is
definitely useful to have a way to capture the robustness of models
against unforeseen attacks, but I wonder whether an average as opposed
to, e.g., a geometric mean, would provide useful insights. In a way, the
average would consider all the attacks equally challenging where, in
reality, this might be the case. (I do actually like the use of mUAR and UAR tho,
as outlined in Section 5.2).

## Additional comments

In Section 4, the authors explain how the distortion size \epsilon_max
is chosen: "[...] the smallest ε which either reduces adversarial
accuracy of an adversarially trained model at distortion size ε below 25
or yields images confusing humans". How is this verified? Do the authors
rely on a user study? Does this only rely on Gilmer et al. (2018)?

[1] https://s2lab.kcl.ac.uk/projects/intriguing/ (IEEE S&P 2020)

---

> ### Author Response · Authors · 2020-11-11
> **Clarification on benefits of ImageNet-UA**
>
> We thank the reviewer for their review and comments.  The main concern seems to be about the benefit of ImageNet-UA in the absence of a theoretical framework for constraining attacks. We first clarify this and then address the other comments.
>
> Though we agree that a theoretical set of constraints on attacks is desirable, because the unforeseen setting admits an extreme diversity of plausible attacks, we are not convinced encapsulating such constraints in a threat model (as is common in computer security) is possible.  Even if a set of constraints claims to characterize valid unforeseen attacks, it is difficult to rule out the existence of a perceptually valid attack which does not lie within them.  As a result, we believe that evaluation of robustness to unforeseen attacks must necessarily have a significant empirical component.
>
> Given this, robust models in practice must be able to generalize to unforeseen attacks whose form is difficult to know or imagine at training time, even those which may lie outside any specific set of constraints.  ImageNet-UA has two roles in the development of such models.
>
> First, when used to evaluate models designed without reference to attacks in ImageNet-UA, it provides a method to demonstrate some level of generalization across attacks.  Some of our attacks are based on common image transformations (JPEG, Snow, Fog) similar to those appearing in practice, while others are a diverse set of perturbations.  Though any specific attack will most likely not appear in practice, the attacks in ImageNet-UA are sufficiently generic that we believe generalizing to them is a necessary condition for a defense to generalize to attacks which do appear in practice.
>
> From this perspective, ImageNet-UA is an initial benchmark to measure progress on generalization across attacks, which our results show is difficult to achieve with known methods. While this type of evaluation against diverse attacks is commonly advocated, it is rare in practice because designing general, easily-optimized, and strong attacks is quite challenging and few are available.  Enabling it is a major contribution of our work.
>
> Second, ImageNet-UA provides a method to stress-test precisely defined threat models.  Namely, we believe that any precise notion of robustness to unforeseen attacks should at least include robustness to the specific diverse attacks in ImageNet-UA.  As an example, evaluation against our attacks provides an empirical test of adversarial training against attacks in such a threat model.
>
> We hope that the reviewer can reevaluate our paper from this viewpoint.  If this allays the reviewer’s concerns about the benefits of ImageNet-UA, we would respectfully ask them to consider improving the rating of this paper.
>
> ## Reasoning about properties or transformations in attacks
> We thank the reviewer for bringing [1] to our attention. Although as discussed above we do not believe it applies in our setting, it offers a quite interesting perspective which we will discuss in a revised version of our paper.
>
> ## Type of mean used in mUAR
> We define mUAR as a simple average over attacks to report a summary statistic, as is common in contemporary multi-task benchmarks (e.g., ImageNet-C, SuperGLUE) which do not have a clear weighting between tasks.  Though we considered this possibility, we did not find a clear ranking of our attacks by strength which remained consistent across defenses, making it difficult to justify any particular weighting.
>
> ## Choice of maximum eps
> This was verified by manual inspection of the resulting attacked images by the authors.  We note that our qualitative results are not very sensitive to the exact choice of eps_max because ATAs are relatively stable in eps_max for attacks where this applied (L1, Fog, and Snow) and adversarial accuracies of most defenses are quite low at eps_max.

---

### Author Response · Authors · 2020-11-13
**Revised version of paper**

We have uploaded a revised version of our paper incorporating the comments from all reviewers and making the changes promised in our responses to the reviews. A particular highlight is the more detailed description of our new attacks in Section 3.1.  Please note also that an anonymized open-source version of our attacks and benchmarking code is available at the link [1] in the introduction.

We respectfully ask the reviewers to take a look at the revised paper when considering our responses.

[1] https://github.com/anon-submission-2020/anon-submission-2020

---

### Decision · Program_Chairs · 2021-01-07
**Final Decision**

**Decision:**

Reject

**Comment:**

In this paper, a defense method against test-time adversarial ML attacks is proposed. Unfortunately, it is not clear whether the proposed method is practically useful or not, because the types of attacks assumed in this paper are too simple and heuristic. Also, the position of the proposed method in the vast amount of existing research on adversarial attacks is not clear. Although the proposed method is conceptually interesting, no evidence is provided that the proposed method is significantly superior to existing approaches.